# Cavity-enhanced photon indistinguishability at room temperature and telecom wavelengths

Lukas Husel[1,7], Julian Trapp[1,7], Johannes Scherzer [1], Xiaojian Wu [2], Peng Wang[2], Jacob Fortner[2], Manuel Nutz[3], Thomas Hümmer[3], Borislav Polovnikov [1], Michael Förg [3], David Hunger [4,5] ✉, YuHuang Wang [2] ✉ & Alexander Högele [1,6] ✉

Indistinguishable single photons in the telecom-bandwidth of optical fibers are indispensable for long-distance quantum communication. Solid-state single photon emitters have achieved excellent performance in key benchmarks, however, the demonstration of indistinguishability at room-temperature remains a major challenge. Here, we report room-temperature photon indistinguishability at telecom wavelengths from individual nanotube defects in a fiber-based microcavity operated in the regime of incoherent good cavity-coupling. The efficiency of the coupled system outperforms spectral or temporal filtering, and the photon indistinguishability is increased by more than two orders of magnitude compared to the free-space limit. Our results highlight a promising strategy to attain optimized non-classical light sources.

The capability of two indistinguishable single photons to interfere on a balanced beam splitter and exit jointly on either one of its output ports is a premise to quantum photonic applications[1] such as quantum teleportation[2], quantum computation[3] or quantum optical metrology[4]. Solid-state-based sources of indistinguishable single photons have witnessed tremendous progress in the past decades[5], and among them semiconductor quantum dots stand out as they enable the generation of pure and indistinguishable single photons[6,7] when coupled to optical microcavities[8–10]. However, their operation is so far restricted to cryogenic temperatures and wavelengths in the near-infrared. These limitations motivate alternative platforms operating at ambient conditions and telecom wavelengths to facilitate long-distance quantum communication in optical fibers at reduced loss. Various quantum emitters have proven capable of emitting pure telecom-band single photons at room temperature, including color centers in silicon carbide[11] and gallium nitride[12]. Recently, the realm of such emitters has

been expanded by luminescent nanotube defects (NTDs) in sp³-functionalized single-wall carbon nanotubes[13]. Unlike other emitters, NTDs allow for precise control over the emission wavelength via covalent side-wall chemistry[14–16]. Moreover, carbon nanotubes are straightforward to integrate with gated structures[17], microcavities[18–21] or plasmonic cavities[22]. These properties, combined with high single photon purity[14,22], render NTDs excellent candidates for the development of sources of quantum light.

As common to solid-state quantum emitters, NTDs are subject to strong dephasing at room temperature. As a result, the coherence time $T_2$ of the emitted photons is orders of magnitude smaller than the population lifetime $T_1$. The respective photon indistinguishability, which can be quantified by $T_2/(2T_1)$[23,24], is therefore limited to vanishingly small values. This limitation represents a major challenge in the development of single photon sources based on NTDs and other solid-state quantum emitters. The strategy of reducing $T_1$ to enhance the

¹Fakultät für Physik, Munich Quantum Center, and Center for NanoScience (CeNS), Ludwig-Maximilians-Universität München, Geschwister-Scholl-Platz 1, 80539 München, Germany. ²Department of Chemistry and Biochemistry, University of Maryland, College Park, MD, USA. ³Qlibri GmbH, Maistr. 67, 80337 München, Germany. ⁴Physikalisches Institut, Karlsruhe Institute of Technology, Karlsruhe, Germany. ⁵Institute for Quantum Materials and Technologies (IQMT), Karlsruhe Institute of Technology (KIT), Herrmann-von-Helmholtz Platz 1, 76344 Eggenstein-Leopoldshafen, Germany. ⁶Munich Center for Quantum Science and Technology (MCQST), Schellingstr. 4, 80799 München, Germany. ⁷These authors contributed equally: Lukas Husel, Julian Trapp. ✉e-mail: david.hunger@kit.edu; yhw@umd.edu; alexander.hoegele@lmu.de

photon indistinguishability via Purcell enhancement[6] has been successfully applied to quantum dots and Erbium ions in various cavity geometries[8–10,25–28] as well as to NTDs by coupling to a plasmonic nanocavity[22]. However, all these experiments were operated in the regimes of coherent or incoherent bad cavity coupling[29], where strong dephasing at ambient conditions limits both photon coherence time and Purcell enhancement, and thus all experiments to date crucially relied on operation at cryogenic temperatures with reduced dephasing. Although at ambient conditions spectral or temporal filtering of mainly incoherent photons would increase the photon coherence in principle, it would come at the cost of drastically reduced collection efficiency. Therefore, enhancement of $T_2$ at efficiencies exceeding those attainable through spectral or temporal filtering has remained elusive for quantum emitters subject to strong dephasing.

Here, we demonstrate enhancement of photon indistinguishability for telecom-band single photons from individual NTDs coupled to an optical microcavity. Motivated by a recent theoretical proposal, we operate the NTD-cavity system in the regime of incoherent good cavity coupling[30], where the photon coherence time is determined by the cavity linewidth. By choosing a cavity with a spectrally narrow linewidth, we enhance $T_2$ and thus the photon indistinguishability of the coupled NTD-cavity system. At the same time, the cavity enhances the emission via the Purcell effect, thus yielding simultaneous increase of both indistinguishability and efficiency unattainable by spectral or temporal filtering. As a consequence, the efficiency of our system outperforms spectral or temporal filtering within the same bandwidth by at least a factor of four, with an estimated increase of photon indistinguishability by two orders of magnitude as compared to free-space NTDs. Our results experimentally establish the regime of incoherent good cavity-coupling as a powerful strategy for optimized sources of quantum light.

## Results

The NTDs used in this work, shown schematically in the left panel of Fig. 1a, were obtained by functionalizing (8,6) carbon nanotubes by diazonium reaction[31,32] (see the Methods section for details). The photoluminescence (PL) excitation map of an aqueous suspension with covalently functionalized carbon nanotubes is shown in Fig. 1b,

with an excitation resonance at 718 nm corresponding to the $E_{22}$ transition and emission via $E_{11}$ around 1170 nm, characteristic of (8,6) chiral tubes[33]. The red-shifted emission peak, labeled as $E_{11}^*$ and centered at 1470 nm, corresponds to the luminescence from excitons localized at nanotube side-wall defects with emission wavelength tuned to the telecom S-band[34] by the choice of the functional group, in this case the 3,4,5-trifluoro-2-chlorosulfonyl-aryl group paired with the hydroxy group[32]. For integration in a fiber-based Fabry-Pérot cavity[35] shown schematically in the right panel of Fig. 1a, the nanotubes were dispersed onto a planar macroscopic mirror with a polystyrene layer on top (see the Methods section for details) to ensure optimal coupling near the antinode of the intra-cavity field. Both spectral and spatial overlap between individual NTDs and the fundamental Gaussian cavity mode were optimized by lateral displacement of the macro-mirror and vertical tuning of the fiber-based micro-mirror via piezoelectric actuators. Photons emitted by the NTD-cavity system were coupled into a single mode fiber upon transmission through the planar mirror.

To implement the regime of incoherent NTD-cavity coupling, we employed a distributed Bragg reflector (DBR) mirror coating for spectrally narrow cavity linewidth at the target wavelength of telecom-band emission. Figure 1d shows jointly the ensemble PL spectrum and the cavity finesse obtained from a transfer matrix simulation of the DBR coating. In the cavity, the NTD states were excited resonantly through the $E_{11}$ transition at near-unity DBR mirror transmission and thus independent of the cavity resonance condition. With finesse values on the order of 1000 at the $E_{11}^*$ transition wavelength, the cavity mode provided the primary radiative decay channel for the NTD emission. A combination of long-pass filters was used to suppress the excitation laser and other emission at wavelengths below 1400 nm before detection.

The effect of cavity-coupling on the photonic spectral bandwidth is illustrated in Fig. 1c. At ambient conditions, the spectral width of the NTD emission profile is dominated by pure dephasing at rate $\gamma^*$, with $\Gamma = 2\gamma^*$ on the order of ten nanometers or 10 meV. This is orders of magnitude larger than the experimental cavity linewidth, which was determined as $\kappa = 35.4 \pm 0.1\,\mu eV$ for the lowest accessible longitudinal mode order, corresponding to $61.7 \pm 0.2\,pm$ in the wavelength domain. The small value of $\kappa$ enables operation of our system in the

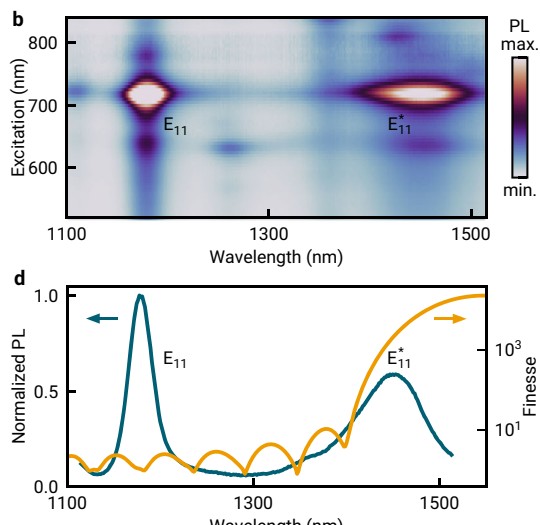

**Fig. 1 | Functionalized carbon nanotubes in an open micro-cavity. a** Schematic of luminescent nanotube defects (NTDs) coupled to the fiber-based open microcavity system with tunable cavity length $L_c$ and lateral displacement degrees of freedom of the macroscopic mirror $x$ and $y$. **b** Photoluminescence (PL) excitation of functionalized (8,6) carbon nanotubes with emission band of fundamental excitons ($E_{11}$) and NTD states ($E_{11}^*$). **c** Schematic spectral weight of strongly dephased free-

space NTD luminescence (dark green) subjected to incoherent cavity coupling (orange). **d** Ensemble PL spectrum (dark green) and cavity finesse in transfer-matrix simulations (orange). The NTD luminescence spectrally close to maximal cavity finesse was excited at the $E_{11}$ transition at near-unity transmission of the cavity mirrors.

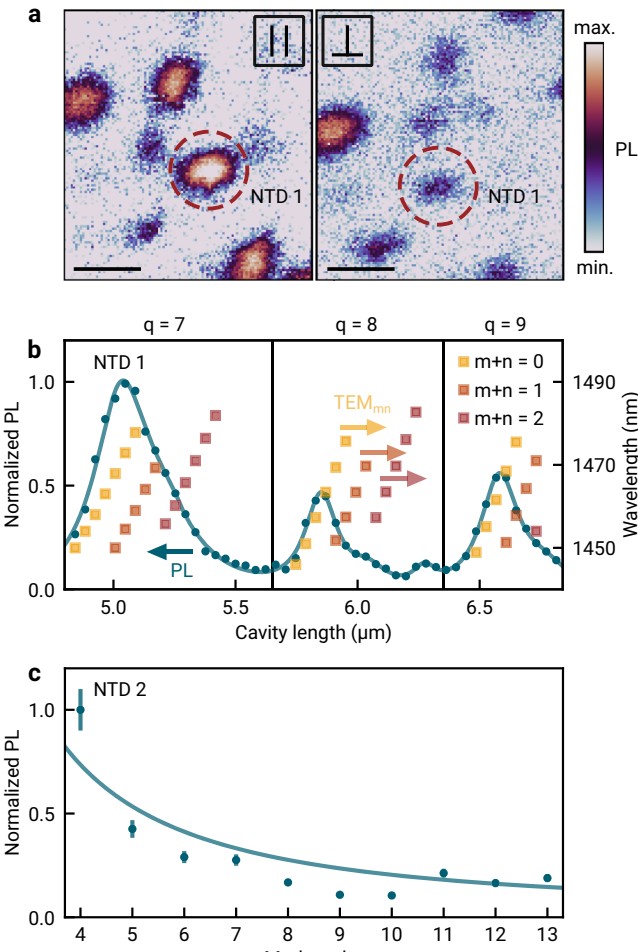

**Fig. 2 | Photoluminescence characteristics of cavity-coupled carbon nanotube defects. a** Cavity-enhanced PL raster-scan maps recorded for two orthogonal linear polarizations. The detection basis is chosen parallel (left) and perpendicular (right) to the axis of the emitter NTD 1 marked by the dashed circle. The scalebar is 5 μm. **b** Normalized PL of NTD 1 as a function of the cavity length, tuned over three longitudinal mode orders (blue circles). The emission spectrum is probed at the resonance wavelengths of the transverse electromagnetic (TEM) cavity modes (yellow, orange and red squares). The solid line was obtained from the fit described in the main text. The colored arrows indicate the respective y-axis. **c** Maximum PL intensity of a different emitter (NTD 2) as a function of the longitudinal mode order, normalized by the coupling efficiency into the single mode fiber. Cavity-enhancement of the PL intensity is inversely proportional to the mode volume $V_c$, as evident from best-fit (solid line). The error bars give the standard uncertainty, dominated by experimental uncertainty in fiber coupling.

regime of incoherent good cavity coupling, where $2g \ll \gamma + \gamma^* + \kappa$ and $\kappa < \gamma + \gamma^*$ holds for the light-matter coupling strength $g$, the population decay rate $\gamma$, $\kappa$ and $\gamma^*$ (see Supplementary Note 1 for details). In this regime, the cavity is incoherently pumped upon initial (incoherent) excitation of the NTD at a rate $R \approx 4g^2/\gamma^*$ [30], which in our system is smaller than the population decay rate. Any photon that is coupled into the resonator will be emitted via the cavity mode on a timescale $1/\kappa$. Since the emission process from the cavity is coherent[30], this constitutes a giant increase in the photon coherence time compared to the free-space limit of $1/\gamma^*$. In the spectral domain, the effect corresponds to a drastic spectral purification as illustrated in Fig. 1c, similar to spectral filtering. This effect is a key feature of the incoherent good cavity coupling regime and is instrumental for enhanced photon indistinguishability.

In addition to the coherence time, the cavity also enhances the emission spectral density, with the enhancement quantified by the Purcell factor $F_p \propto g^2$ [36] (see Supplementary Note 3 for details). Increasing $F_p$ via the light-matter coupling strength $g$ increases the single photon efficiency, i.e. the probability that a photon is emitted into the cavity mode. In the incoherent good cavity regime, this probability is smaller than the free-space quantum yield due to the large mismatch in the spectral bandwidths of the emitter and the cavity. However, as we demonstrate in the following, maximizing $g$ (achieved in our case by minimizing the microcavity mode volume) results in an efficiency which by far exceeds that obtained by filtering at a spectral bandwidth $\kappa$ or an equivalent temporal bandwidth.

Individual NTDs were identified in the cavity from maps of PL intensity as in Fig. 2a, recorded upon lateral raster-scan displacement of the macroscopic mirror for a fixed cavity length. The two maps of Fig. 2a were acquired for two orthogonal linear polarizations in the detection path and feature bright PL spots with lateral extent given by the point spread function of the Gaussian fundamental cavity mode with a waist of 2 μm. The left (right) map in Fig. 2a was obtained for parallel (orthogonal) orientation of the polarization axis with respect to the nanotube with NTD 1. The contrast in the brightness between the two maps for most PL hot-spots indicates a large degree of linear polarization at the emission sites, a hallmark of the well-known antenna effect in individual carbon nanotubes[37,38].

In Fig. 2b, we show the normalized PL intensity of NTD 1 as the cavity length is tuned over three longitudinal mode orders $q = 7, 8$ and 9. For each mode order, we observe an asymmetric emission profile, stemming from higher order transverse electromagnetic (TEM) modes. Since the cavity linewidth $\kappa$ is much smaller than the emitter PL linewidth $\Gamma$, the NTD emission spectrum is probed at the resonance wavelength of each TEM-mode[39] with resonance wavelengths given explicitly on the right axis of Fig. 2b. We fitted the data by the sum of three Lorentzians for each longitudinal mode order, with the result shown as the solid line in Fig. 2b (TEM$_{mn}$ mode orders with $n + m > 2$ were neglected due to vanishing contributions). From the fit, we obtained the emission wavelength $1465 \pm 3$ nm, and a FWHM linewidth of $28 \pm 5$ nm, corresponding to $\gamma^* = 8 \pm 2$ meV.

Figure 2c shows the effect of the cavity mode volume on the photon emission efficiency. We measured the collected PL intensity of a different NTD with comparable brightness for ten consecutive longitudinal mode orders, normalized to the largest value and corrected for the variation of the measured fiber coupling. The fiber coupling efficiency depends on the mode waist, which in turn changes with cavity length. We observed an increase in the PL intensity by a factor of six as the cavity was tuned to the lowest accessible longitudinal mode order $q = 4$. This mode order corresponds to an inter-mirror separation of 2.6 μm, mainly limited by the profile depth of the fiber mirror of 2 μm. The increase in the PL intensity stems from an enhancement in light-matter coupling strength $g$ as the cavity length and hence the mode volume $V_c$ is decreased. For our regime of low Purcell enhancement, where the NTD population lifetime is mainly unaffected by the cavity, the emission intensity is proportional to $g^2$, which in turn is inversely proportional to $V_c$ (see Supplementary Note 3 for details). A fit of $\alpha V_c^{-1}$, with $V_c$ calculated from the cavity length $L_c = q\lambda/2$ [35] and the amplitude $\alpha$ as a free fit parameter, yields a good correspondence with the data (solid line in Fig. 2c).

Operating the coupled NTD-cavity system at maximum cavity-enhancement of the PL intensity, we determined second-order correlations in photon emission events with a fiber-based Hanbury–Brown–Twiss (HBT) setup shown schematically in Fig. 3a. Photons generated via pulsed laser excitation were coupled into a fiber beamsplitter, and detection events at the output ports were time-correlated to obtain the normalized second-order auto-correlation function $g^{(2)}_{HBT}(\tau)$. The shot-noise limited results of the

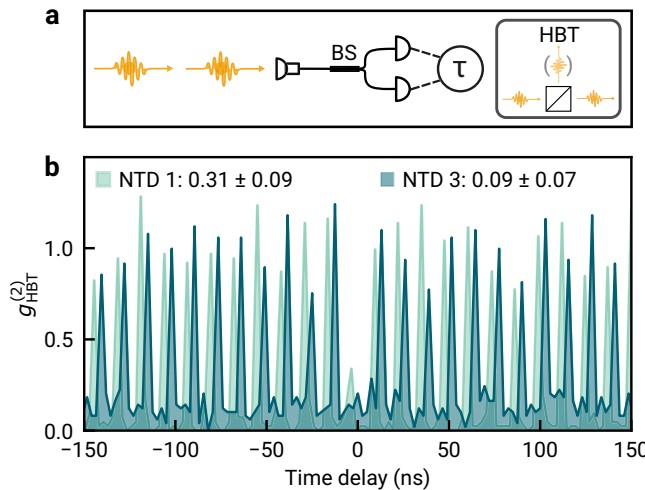

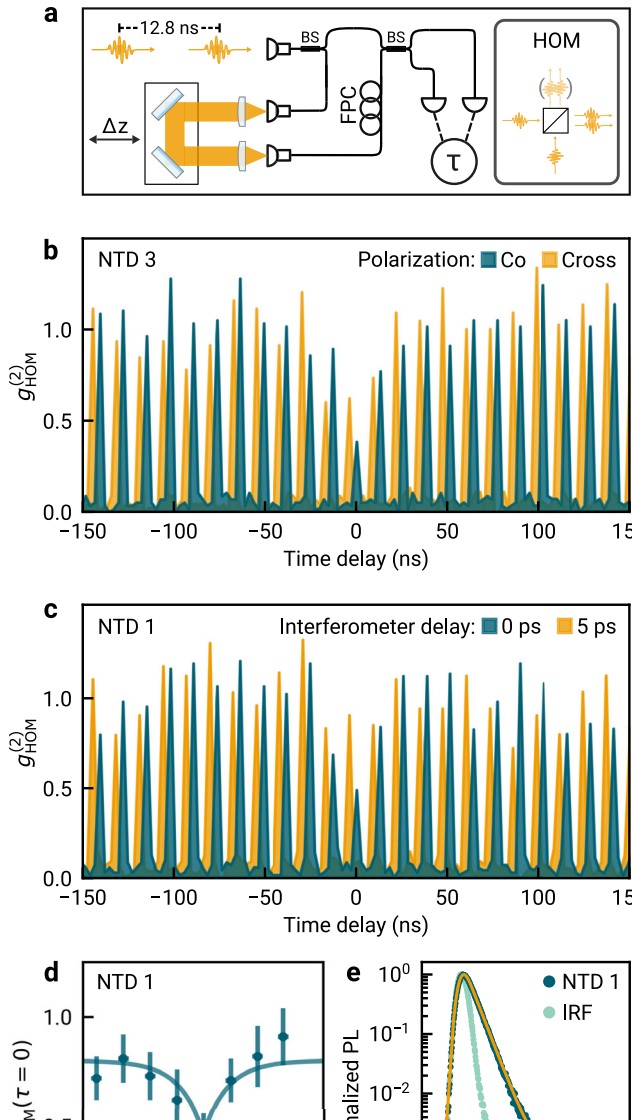

**Fig. 3 | Telecom-band room-temperature single photons from the coupled NTD-cavity system. a** Schematic of a Hanbury–Brown–Twiss (HBT) setup based on a fiber beamsplitter (BS). **b** HBT autocorrelation function of cavity-coupled NTD 1 (light green) and NTD 3 (dark green), with second order coherence at zero time delay $g^{(2)}_{HBT}(0) = 0.31 \pm 0.09$ and $0.09 \pm 0.07$, respectively.

HBT experiment on two distinct NTDs are shown in Fig. 3b, with the corresponding antibunching values $g^{(2)}_{HBT}(0) = 0.31 \pm 0.09$ and $0.09 \pm 0.07$ as measures of the single photon purity.

The photon indistinguishability was quantified in Houng-Ou-Mandel (HOM) type experiments using an imbalanced Mach–Zehnder interferometer shown schematically in Fig. 4a. The train of single photon pulses generated by the source was first split in a fiber beamsplitter. The time delay $\Delta t$ in the interferometer was tuned by the path difference $\Delta z$ with an adjustable delay stage to enable two-photon interference between consecutively emitted photons at the second beam splitter. In this setting, a delay of zero implies a separation by one excitation pulse. The relative polarization between the interferometer arms was set by fiber polarization controllers, and the detection events at the output ports were time-correlated to obtain the HOM-autocorrelation function $g^{(2)}_{HOM}(\tau)$ (see the Methods section for details). First, we initialized the interferometer at zero delay and performed a two-photon interference experiment for co- and cross-polarized interferometer arms on NTD 3. The shot-noise limited results are shown in Fig. 4b. For the co-polarized configuration, we observe a reduction of the measured correlations at zero time delay. This is a hallmark of quantum coherent two-photon interference: the (partially) indistinguishable single photons arriving simultaneously at different input ports of the beamsplitter are likely to exit at the same output port, resulting in reduced correlations at zero time delay[8,9,40]. We quantify the respective degree of the photon indistinguishability by the two-photon interference visibility $v$ that one would detect in an interferometer with balanced beamsplitters and unity classical visibility[10]. We obtain $v = 0.51 \pm 0.21$ for the data in Fig. 4b, taking into account non-identical reflection and transmission of the beamsplitters and finite single photon purity of NTD 3 (see Supplementary Note 4 for details).

Successively, we performed the HOM interference experiment for varying interferometer delays on NTD 1, with autocorrelation histograms for interferometer delays of 0 and 5 ps shown in Fig. 4c. The observed reduction in correlations at zero time delay is again a hallmark of two-photon interference, where tuning between the two interferometer delay settings is approximately equivalent to switching the polarization configuration as in Fig. 4b. In Fig. 4d, we show the measured value of the HOM autocorrelation function at zero time delay for varying interferometer delay. Upon transition through zero-delay, we observed the characteristic HOM dip due to reduced cross-

**Fig. 4 | Demonstration of cavity-enhanced photon indistinguishability. a** Schematic of the imbalanced Mach–Zehnder interferometer to probe the photon indistinguishability in Hong-Ou-Mandel (HOM) type experiments based on fiber beamsplitters (BS). The time delay between the interferometer arms was tuned via the displacement $\Delta z$, and their relative polarization by the fiber polarization controller (FPC) in one arm. **b** HOM autocorrelation function of NTD 3 for co-polarized (dark green) and cross-polarized (orange) interferometer arms with delay of one excitation pulse. The difference in the coincidence probabilities at zero-delay is a hallmark of two-photon interference with visibility $v = 0.51 \pm 0.21$. **c** HOM autocorrelation function of NTD 1, measured in co-polarized interferometer configuration for interferometer delays 0 ps (dark green) and 5 ps (orange). Zero interferometer delay again corresponds to delay by one excitation pulse separation. **d** HOM autocorrelation function at time delay $\tau = 0$ for NTD 1 as a function of the interferometer delay, with visibility $v = 0.65 \pm 0.24$. The solid line is an empirical fit to the HOM dip described in the main text. The horizontal error bars correspond to the standard uncertainty in the interferometer delay; the vertical error bars correspond to the standard uncertainty determined as described in the Methods section. **e** Temporal PL decay of NTD 1 (dark green data) and instrument response (light green data). The orange line shows the result of a biexponential decay model.

channel correlations by two-photon interference, described by the empirical formula $c[1 - a\exp(-|\Delta t|/\tau_{HOM})]$, where $a$ is an amplitude, $c$ is an offset at large interferometer delays $\Delta t$, and $\tau_{HOM}$ is the characteristic timescale of the HOM interference[40]. From the best fit to the data shown by the solid line in Fig. 4d, we determined $\tau_{HOM} = 2 \pm 2$ ps, and a visibility of $0.65 \pm 0.24$ (see Supplementary Note 4 for details), consistent with the value of $0.51 \pm 0.21$ for NTD 3.

The characteristic two-photon interference time scale $\tau_{HOM}$ is given by the jitter in the photon arrival time at the beamsplitter, which in turn is determined by the population lifetime[23] (see Supplementary Note 5 for details). For the emitter NTD 1, the fit to the data in Fig. 4d thus implies a population decay within a few picoseconds. This time scale can be associated with the short decay component of the biexponential PL decay characteristic for NTDs[14,41]. The fast and slow decay channels with time constants $\tau_{fast}$ and $\tau_{slow}$ arise from an interplay of bright and dark exciton reservoirs, with $\tau_{fast}$ as short as a few picoseconds and relative decay amplitudes close to unity in larger-diameter nanotubes[41]. In Fig. 4e, we show by the solid line the result of a cavity-coupled biexponential model decay with $\tau_{fast} = 2$ ps and $\tau_{slow} = 91$ ps, convoluted with the instrument response function, together with the measured PL decay for NTD 1.

Although the short decay component is not resolved directly in the instrument-response limited data of Fig. 4e, the identification of $\tau_{fast}$ with $\tau_{HOM}$ is plausible. In the framework of the incoherent good cavity regime, the feeding of the cavity through the fast decay channel generates photons with near-unity visibility[30]. The actual visibility in Fig. 4d is lower than unity ($0.65 \pm 0.24$), most probably due to photons generated via the slow process with lifetimes exceeding the cavity coherence time of 20 ps, which renders them partly distinguishable. A reduction in visibility is also backed by our model for time-dependent NTD-cavity coupling, which predicts $v = 0.3$ for the NTD 1 in Fig. 4d (see Supplementary Note 4 for details). The deviation between measured and estimated value is consistent with operation of our experiment at wavelengths on the edge of the DBR stopband (see Fig. 1c). In this regime, small shifts towards larger resonance wavelength caused by cavity length drifts can decrease the cavity linewidth by a factor of up to two and in turn result in increased visibility, which is inversely proportional to $\kappa$[30].

The visibility in the two-photon interference data in Fig. 4d corresponds to a 217-fold enhancement of the value estimated for the free-space limit (see Supplementary Note 4 for details). For spectrally filtered free-space emission, the same visibility can be achieved in principle, yet at the cost of very low single photon efficiency. In the incoherent good cavity regime implemented here, the measured lower bound $\min(\beta_c) = (4.0 \pm 0.1) \cdot 10^{-3}$ and expected value $\beta_c = 6.6 \cdot 10^{-3}$ for the Purcell-enhanced single photon efficiency are a factor of four and seven larger than the estimated upper bound $\beta_{fs} = \kappa/(\pi\gamma^*) = 1 \cdot 10^{-3}$ for spectrally filtered free-space decay, whose actual value we expect to be at least one order of magnitude smaller when taking into account the non-unity NTD quantum yield (see Supplementary Note 3 for details). Further benefit arises from the fiber-based design of our cavity, which in principle allows unity in-fiber coupling efficiency in contrast to free-space collection with inherent diffraction losses.

## Discussion

To conclude, we have presented a room-temperature source of telecom-band single photons with emission efficiency and indistinguishability drastically enhanced by incoherent NTD-cavity coupling. To our knowledge, our results represent the first demonstration of cavity-enhanced indistinguishability for a quantum emitter with room-temperature dephasing. We estimate that the current two-photon interference visibility of about 0.5 can be improved to near-unity values by increasing the cavity finesse to 35,000, a feasible value with open fiber cavites[20]. Simultaneously, a further reduction of the mode volume to recently reported values[42] would yield an enhancement in emission efficiency by another order of magnitude. Even without these improvements, our results represent a major step towards room-temperature quantum photonic devices for applications at telecom-wavelength in optical quantum computation[43] or long-distance communication relying on optical quantum repeaters[44].

## Methods
### Sample preparation
The NTDs were prepared by functionalizing (8,6) carbon nanotubes based on a method we reported previously[32]. Briefly, raw HiPco SWCNT material (NoPo Nanotechnologies, India) was dissolved in chlorosulfonic acid (99%, Sigma-Aldrich) at a concentration of 0.5 mg/mL, followed by adding 2-amino-4,5,6-trifluorobenzen-1-sulfonyl chloride, which was synthesized from 3,4,5-trifluoeoaniline, and NaNO2 (ReagentPlus® >99.0%, Sigma-Aldrich) to concentrations of 0.24 mg/mL and 0.2 mg/mL, respectively. After fully mixed, the acid mixture was then added drop-by-drop to Nanopure® water with vigorous stirring, resulting in the formation of NTD functionalized carbon nanotubes that precipitated from the solution as black precipitates. The precipitates were filtered and rinsed with an excessive amount of Nanopure® water. The synthesized NTDs were dissolved in 2% (wt/v) sodium deoxycholate (DOC, Sigma-Aldrich, ≥97%) solution and centrifuged at 16400 rpm for 1 h to remove any bundles. The nanotubes with NTDs were then sorted by aqueous two-phase extraction[32,45] in a solution of 2% (w/v) DOC in deuterium oxide (D2O, Cambridge Isotope Laboratories, Inc. 99.8%) to obtain NTDs on (8,6) chirality enriched nanotubes.

Next, a macroscopic planar mirror was spin-coated with a 10 μL solution of 3% (wt/v) polystyrene/toluene, at 2000 RPM for 1 min, resulting in the formation of a polystyrene spacer layer estimated to be 150-nm thick. The coated mirror was then vacuum-dried at room temperature for 24 h before being deposited with 5 μL of the NTDs containing solution by spin-coating at 3000 RPM for 1 min.

### Fiber-based cavity
The experiments were conducted in an ultra-stable fiber-based open-cavity platform (*Qlibri Quantum*, Qlibri GmbH). The cavity is formed by a microscopic concave fiber mirror with a radius of curvature of 25 μm, fabricated by CO$_2$ laser ablation[35,46], and a macroscopic planar mirror with a 150-nm thick polystyrene spacer layer and functionalized carbon nanotubes on top. The spacer layer was included to place NTDs close to an antinode of the intra-cavity field. Three translational degrees of freedom are accessible through piezoelectric positioners, allowing for lateral scans and length-tuning of the cavity with sub-nanometer precision. Fiber and sample mirror have identical DBR coatings, designed for high reflectivity at telecom wavelengths (minimum transmission $T = 95.2$ ppm at wavelength of 1535.4 nm) and fabricated by ion beam sputtering (Laseroptik GmbH). At a wavelength of 1468 nm, close to the $E_{11}^*$ peak maximum, the largest measured finesse was $3010 \pm 10$ for the lowest accessible longitudinal mode order $q = 4$. For this mode order, corresponding to a mirror distance of $L_c = 2.6$ μm, we calculated a mode waist of $\omega_0 = 2$ μm and a cavity mode volume of $V_c = 8.2$ μm$^3$[35].

### Photoluminescence and photon correlation experiments
PL measurements were performed under resonant excitation of the $E_{11}$ transition using a pulsed supercontinuum white light source (NKT SuperK Extreme) at a repetition rate of 78 MHz that was spectrally filtered in a home-built monochromator to a linewidth of 2 nm. The cavity was tuned on resonance with the $E_{11}^*$ transition by changing the mirror distance. The PL emitted through the planar mirror of the cavity was collimated by an achromatic doublet lens (Thorlabs AC127-019-C-ML), filtered with two longpass filters (Thorlabs FEL1400, band edge 1400 nm, and Semrock BLP02-1319R-25, band edge 1320 nm) and coupled into a single mode fiber. Detection was performed with a pair

of superconducting nanowire single photon detectors (Scontel TCOPRS-CCR-SW-85) and time-correlated with a TCSPC module (Swabian Instruments Time Tagger Ultra and PicoQuant Pico-Harp300). Second-order photon correlation measurements were performed in a standard Hanbury–Brown–Twiss configuration. For Hong-Ou-Mandel type two-photon interference experiments, a home-built fiber-based imbalanced Mach–Zehnder interferometer was employed. A mechanical delay stage was used to tune the interferometer delay on sub-picosecond scale. Polarization was set by fiber-polarization controllers (Thorlabs FPC562).

Photon correlation histograms were obtained by integrating detection events in 2.5 ns wide windows. The resulting histograms feature prominent peaks separated by the delay between the excitation pulses. To obtain the correlation functions $g^{(2)}_{\mathrm{HBT}}$ and $g^{(2)}_{\mathrm{HOM}}$, we normalized the histograms with respect to the average height of histogram peaks $N_\infty$ at large time delays. The standard uncertainty of the measured peak height $N_0$ at $\tau = 0$ is given by $\sqrt{N_0}$[47] and is the dominant uncertainty in the measurement of $N_0$. The standard uncertainty in quantities derived from measured peak heights was obtained by Gaussian error propagation, considering the uncertainties in all input parameters. The normalized second-order correlation at zero time delay $g^{(2)}(0)$ was obtained from the measured histograms as $g^{(2)}(0) = N_0 / N_\infty (1 \pm 1/\sqrt{N_0})$[47] including dark count and background correction[8]. The uncertainties in $N_\infty$ and background were found to have negligible influence on this measurement, whose uncertainty is dominated by the uncertainty in $N_0$.

## Data availability

The source data generated in this study have been deposited in the LMU Open Data database under accession code https://doi.org/10.5282/ubm/data.460.

## Code availability

The codes that support the findings of this study are available from the corresponding authors upon request.

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

## Acknowledgements

We gratefully acknowledge helpful discussions with Lukas Knips and support by Max Huber for manufacturing of the cavity. This research was funded by the European Research Council (ERC) under the Grant Agreement No. 772195 as well as the Deutsche Forschungsgemeinschaft (DFG, German Research Foundation) within the Germany's Excellence Strategy EXC-2111-390814868. L.H. and A.H. acknowledge funding by the Bavarian Hightech Agenda within the EQAP project. B.P. acknowledges funding by IMPRS-QST. D.H. acknowledges support by the Karlsruhe School of Optics & Photonics (KSOP). Y.W. gratefully acknowledges the U.S. National Science Foundation for funding support (grant no. PHY1839165 and CHE2204202).

## Author contributions

D.H., Y.W. and A.H. conceived the project. L.H. and J.T. set up and performed the experiments with contributions by J. S., evaluated the data, and carried out theoretical analysis and modeling. X.W. led the sample preparation of defect-tailored carbon nanotubes synthesized by P.W. with contributions from J.F. and supervision by Y.W. B.P. contributed to the initial sample characterization by optical spectroscopy. M.N., T.H. and M.F. designed and manufactured the cavity and provided support for its operation. L.H., J.T., D.H. and A.H. analyzed the data. L.H., J.T. and A.H. wrote the manuscript. All authors commented on the manuscript.

## Funding

## Competing interests

The authors declare no competing interests.
