## [Peer Review File · Nature Communications]

Cavity-enhanced photon indistinguishability at room temperature and telecom wavelengthsREVIEWER COMMENTS

Reviewer #1 (Remarks to the Author):

The authors discuss a Hong Ou Mandel experiment, conducted at room temperature on a nanotube quantum emitter which is embedded in a microcavity of high quality factor (finesse >1000). They claim significant quantum coherence, in a regime of good cavity incoherent coupling. With more convincing results, I would rate the paper as highly relevant.

However, unfortunately, I do not find the results convincing.

a) I find it very odd and rather reader-unfriendly, that the authors first discuss the HOM histogram chart of emitter '3' in Fig 4b, and then switch to emitter '1' in Fig 4c. ND1 has such a high $g_2(0)$ value, that it is probably questionable whether the outcome of a HOM study can provide a solid result.

b) My main concern, however, is the assessment of the indistinguishability values in general. Both, in fig 4b, as well as in S1, the difference between the central peak in the histogram is on the order of the fluctuations of the neighbouring peaks (most notably in fig 4b, the two neighboring peaks on the negative time delay axis). More concerning, due to only 50 (or less) binned coincidence counts, the fluctuation of the correlation peak height is on the order of the signal itself. I really don't think that such a small statistics is sufficient to draw strong conclusions. It seems that the authors miss one order of magnitude of signal bins in their correlations.

To conclude, due to the small correlation coincidences, I cannot accept the paper for publication. If the authors can engineer the cavity to increase the count rate, allowing them to accumulate a larger statistics, I will be happy to revise my opinion.

Reviewer #2 (Remarks to the Author):

The authors show experimentally, that coupling a single-photon emitter to a single mode microcavity in the regime of incoherent good coupling allows to achieve highly indistinguishable and relatively bright single photon emission at room temperature and telecom wavelength. This is by no doubt a very promising experimental achievement which

would have a substantial impact on the field of solid state quantum emitters. While, the mechanism of the cavity enhanced photon indistinguishability is generally known (described for e.g. in [9]), this work demonstrates for the first time implementation of this approach at room temperatures and telecom wavelengths.

I certainly recommend the manuscript for publication. I however have a minor request:

I believe, authors could dedicate some space in the main text to briefly explain the origin of the enhanced photon indistinguishability. While the Supp Info contains kinetic equations which by tuning the parameters reproduce the experimental data, I believe an explanation would be very helpful and would increase the readability dramatically.

Also there is a typo in Eq. 5 in Supp Info. ρ_{db} should be ρ_d

Reviewer #3 (Remarks to the Author):

The authors demonstrate that the photon indistinguishability be improved in the regime of incoherent good cavity coupling. Furthermore, the efficiency of the coupled system is better than the spectral or temporal filtering. They show the photon indistinguishability is increased by 2 orders of magnitude compared to free space limit. The work looks promising; however, I have a few questions and suggestions for the improvement of the presentation of the work. Authors demonstrate Purcell enhanced indistinguishable single photon.

1. For better understanding of the Cavity enhanced generation of indistinguishable photons Author should cite some more recent papers on Purcell enhanced generation of indistinguishable single photons (Purcell-Enhanced and Indistinguishable Single-Photon Generation from Quantum Dots Coupled to On-Chip Integrated Ring Resonators- Łukasz Dusanowski et al, Nano Lett. 2020, 20, 9, 6357–6363, <https://doi.org/10.1021/acs.nanolett.0c01771>; Deterministic coupling of quantum emitter to surface plasmon polaritons, Purcell enhanced generation of indistinguishable single photons and quantum information processing, Lakshminarayan Sharma et al Optics Communications, 2021, 127139 <https://doi.org/10.1016/j.optcom.2021.127139> and High Purcell factor generation of indistinguishable on-chip single photons, Feng Liu, Nature Nanotechnology volume 13, pages835–840 (2018)

2. Authors claim that the $v = T_2/2T_1$ refers to two photon interference visibility. However, the cited reference mentions it as efficiency of photon coalescence. How are these two related?

3. Results from different type of cavities (For example, bad cavity) for generation of Purcell enhanced indistinguishable photon is missing.

4. For the sake of reproducibility of results, Steps for Fabrication of cavity including DBR coating preparation should be elaborated.

Telecom-band single photon source with cavity-enhanced indistinguishability at room temperature

by Lukas Husel et al.

Responses to the Reviewers

We thank all Reviewers for taking the time and effort to critically assess our manuscript. We feel that the feedback provided has helped us to greatly improve our work. Below, we address the reviews in point-by-point responses.

We note that we now use NTD instead of ND as abbreviation for luminescent nanotube defects to avoid potential confusion with emitters in nanodiamond.

Reviewer #1

The authors discuss a Hong Ou Mandel experiment, conducted at room temperature on a nanotube quantum emitter which is embedded in a microcavity of high quality factor (finesse >1000). They claim significant quantum coherence, in a regime of good cavity incoherent coupling. With more convincing results, I would rate the paper as highly relevant. However, unfortunately, I do not find the results convincing.

We thank the Reviewer for the critical assessment of our work, and believe to be able in the following to convince the Reviewer to support our work as highly relevant. The main skepticism of the Reviewer refers to the signal-to-noise ratio in our data, which we address in an elaborate and quantitative discussion leaving no room for doubts on the statistical significance of our data and the integrity of our interpretation.

a) I find it very odd and rather reader-unfriendly, that the authors first discuss the HOM histogram chart of emitter '3' in Fig 4b, and then switch to emitter '1' in Fig 4c. ND1 has such a high $g_2(0)$ value, that it is probably questionable whether the outcome of a HOM study can provide a solid result.

We begin by pointing out that the value of $g_{HBT}^{(2)}(0) = 0.31 \pm 0.09$ determined for NTD1 is sufficient to support meaningful HOM-measurements. As the value within error bars is below 0.5, the experiment is clearly performed on a quantum emitter. Since the value is above 0, the probability of emitting two photons in one excitation cycle is finite, and the second "noise" photon will result in a coincidence detection event at electronic delay $\tau = 0$ with finite probability. This probability, however, is independent of the interferometer delay. As a result, nonzero $g_{HBT}^{(2)}(0)$ simply offsets the histogram peak corresponding to $g_{HOM}^{(2)}(0)$ without creating additional peaks or dips. To illustrate this, we plot in the following figure the actually measured values together with a hypothetical $g_{HOM}^{(2)}(0)$ for a perfect single photon emitter:

Figure R1: HOM measurement for NTD1. Measured $g_{HOM}^{(2)}(0)$ (dark blue dots), fit to the HOM dip (dark blue line), hypothetical $g_{HOM}^{(2)}(0)$ for perfectly distinguishable photons emitted by NTD1 (black dashed line), hypothetical $g_{HOM}^{(2)}(0)$ for a perfect single photon emitter ($g_{HBT}^{(2)}(0) = 0$, light green line).

Irrespective of the value of $g_{HBT}^{(2)}(0)$, if two single photons impinge on the second beamsplitter of the HOM setup with a temporal distance smaller than their cavity-enhanced coherence time, they will coalesce into the same output port, which reduces the coincidence probability. As a result, we observe a pronounced HOM dip upon tuning the interferometer delay, a well-established hallmark of two-photon interference. We note that two-photon interference for emitters with $g_{HBT}^{(2)}(0)$ values similar to that of NTD1 has been recently reported for a cryogenic quantum dot (Ollivier et al., Phys. Rev. Lett. 126, 063602, 2021).

Our shot-noise limited HOM histograms exhibit sufficiently large coincidence peak heights to unambiguously support our interpretation. The standard uncertainty in our measured peak heights N is given by \sqrt{N} (see Ref. [46] of the revised main text for a derivation), and we rigorously quantify the uncertainty in quantities derived from measured peak heights by Gaussian error propagation, considering the uncertainties in all input parameters. This is standard scientific practice, with the explicit calculation given in previous works (e.g. Delteil et al., Nat. Mater. 18, 219, 2019, Müller et al., Optica 3, 931, 2016, and Luo et al., Nano Lett. 19, 9037, 2019). In our work, we obtain $V = 0.65 \pm 0.24$ for the two-photon interference visibility of NTD1. The probability that the photons exhibit a statistically significant degree of quantum coherence (i.e. that the value of V is larger than zero) is 99.6%, providing clear evidence for two-photon interference. The probability that V is two orders of magnitude larger than the expected free-space value is 93%, establishing a statistically significant proof for giant cavity-enhancement of the indistinguishability. Since the statistical significance of the HOM measurements on NTD3 and NTD1 are equivalent (see our response to point b) below), we decided to refrain from reversing the order of the two emitters in the discussion around Fig. 4. In the revised Methods section, we now explicitly state that the uncertainty in the quantities derived from the measurements of N was computed as described above.

b) My main concern, however, is the assessment of the indistinguishability values in general. Both, in fig 4b, as well as in S1, the difference between the central peak in the histogram is on the order of the fluctuations of the neighboring peaks (most notably in fig 4b, the two neighboring peaks on the negative time delay axis). More concerning, due to only 50 (or less)

binned coincidence counts, the fluctuation of the correlation peak height is on the order of the signal itself. I really don't think that such a small statistics is sufficient to draw strong conclusions. It seems that the authors miss one order of magnitude of signal bins in their correlations.

We respectfully disagree with the Reviewers' conclusion that the mean peak height (coincidence count number) in the HOM histograms recorded for the emitter NTD3 is too small to support evidence for two-photon interference. The probability that statistical fluctuations induce a large absolute difference between the coincidence peaks in co- and cross-polarized interferometer configuration indeed decreases with increasing mean coincidence count number. Irrespective of this obvious fact, the uncertainty in any individual measured peak height N is determined by \sqrt{N} . Quantifying the uncertainties in the measured peak heights and other measured quantities rigorously by following the established practice outlined above, we find $V = 0.51 \pm 0.21$ for NTD3. The probability that the photons exhibit a statistically significant degree of quantum coherence (i.e. that the value of V is larger than zero) is again 99.6%, providing further evidence for two-photon interference in our experiments. We note that the peaks at time delay $\tau = -12.5$ ns, on which the reviewer has placed particular emphasis, fluctuate around the expected value of 0.77 ± 0.02 within statistically significant bounds (see the revised Supplementary Information for details). We also note that recent experiments have demonstrated two-photon interference from cryogenic quantum emitters at signal-to-noise ratios comparable to or worse than those in our experiments (Fournier et al., Phys. Rev. Applied 19, L041003, 2023 and Luo et al., Nano Lett. 19, 9037, 2019, with as little as 20 mean coincidence counts as detailed in the supplement thereof).

To conclude, due to the small correlation coincidences, I cannot accept the paper for publication. If the authors can engineer the cavity to increase the count rate, allowing them to accumulate a larger statistics, I will be happy to revise my opinion.

Our quantitative analysis of experimental uncertainties, consistent with recent results from related experiments, leaves no room for doubt that the signal-to-noise ratio in our HOM measurements on both NTDs 1 and 3 is sufficient to support our results and conclusions with very high statistical significance. With the established procedure described, and the relevant numbers provided, we are confident to have resolved the skepticism of the Reviewer without the need of additional experiments or engineered cavities. We also anticipate that the Reviewer will find our elaborate, factual and quantitative arguments convincing, and will support the publication of our work in Nature Communications.

Reviewer #2

The authors show experimentally, that coupling a single-photon emitter to a single mode microcavity in the regime of incoherent good coupling allows to achieve highly indistinguishable and relatively bright single photon emission at room temperature and telecom wavelength. This is by no doubt a very promising experimental achievement which would have a substantial impact on the field of solid state quantum emitters. While, the mechanism of the cavity enhanced photon indistinguishability is generally known (described for e.g. in [9]), this

work demonstrates for the first time implementation of this approach at room temperatures and telecom wavelengths.

I certainly recommend the manuscript for publication.

We thank the Reviewer for the positive assessment of our work, for rating our experimental implementation as very promising and substantially impactful, and for recommending our work for publication in Nature Communications.

I however have a minor request:

I believe, authors could dedicate some space in the main text to briefly explain the origin of the enhanced photon indistinguishability. While the Supp Info contains kinetic equations which by tuning the parameters reproduce the experimental data, I believe an explanation would be very helpful and would increase the readability dramatically.

We thank the Reviewer for this constructive suggestion. In response, we have extended in the last two paragraphs of page 4 the discussion of spectral purification illustrated in Fig. 1c to the time domain, and added an explanation for the origin of Purcell enhanced emission efficiency in the incoherent regime of cavity coupling following the theoretical analysis of Auffèves et al. (Phys. Rev. A 79, 053838, 2009). In our experiments, the time domain picture of enhanced photon coherence applies to both components in the biexponential photoluminescence decay as in Fig. 4d, and we hope that this complementary perspective will add to the understanding of the origin of the HOM-timescale described in the main text. The above-mentioned Purcell enhancement of photon emission efficiency is also demonstrated by the increase in the PL intensity in Fig. 2c, which we now point out when discussing this figure. We believe that these additions motivated by the Reviewer have substantially improved the context for the presentation of our work.

Also there is a typo in Eq. 5 in Supp Info. ρ_{db} should be ρ_d

We thank the Reviewer for pointing out the typo, which we have corrected.

Reviewer #3

The authors demonstrate that the photon indistinguishability be improved in the regime of incoherent good cavity coupling. Furthermore, the efficiency of the coupled system is better than the spectral or temporal filtering. They show the photon indistinguishability is increased by 2 orders of magnitude compared to free space limit. The work looks promising; however, I have a few questions and suggestions for the improvement of the presentation of the work. Authors demonstrate Purcell enhanced indistinguishable single photon.

We thank the Reviewer for the positive assessment of our work and the constructive questions and suggestions for the improvement of its presentation.

1. For better understanding of the Cavity enhanced generation of indistinguishable photons Author should cite some more recent papers on Purcell enhanced generation of indistinguishable single photons (Purcell-Enhanced and Indistinguishable Single-Photon Generation from Quantum Dots Coupled to On-Chip Integrated Ring Resonators- Łukasz

Dusanowski et al, Nano Lett. 2020, 20, 9, 6357–6363, <https://doi.org/10.1021/acs.nanolett.0c01771>; Deterministic coupling of quantum emitter to surface plasmon polaritons, Purcell enhanced generation of indistinguishable single photons and quantum information processing, Lakshminarayan Sharma et al Optics Communications, 2021, 127139 <https://doi.org/10.1016/j.optcom.2021.127139> and High Purcell factor generation of indistinguishable on-chip single photons, Feng Liu, Nature Nanotechnology volume 13, pages835–840 (2018)

We thank the Reviewer for pointing out these relevant references, which we now cite in the introduction. We believe that the references indeed help to illustrate the strategy of Purcell-enhancing the population lifetime in order to achieve large indistinguishability in different regimes of cavity-coupling and cavity geometries at cryogenic temperatures, see our elaborate response to point 3 for details.

2. Authors claim that the $v = T2/2T1$ refers to two photon interference visibility. However, the cited reference mentions it as efficiency of photon coalescence. How are these two related?

We thank the Reviewer for pointing out this discrepancy. The existing literature is somewhat ambiguous in labelling measurable quantities in a HOM experiment. The following statement is however supported across the literature: If two single photons emitted by a two-level system impinge on opposite input ports of a 50:50 beamsplitter, the probability that they coalesce into the same output port is given by $T2/(2T1)$. Some references (e.g. Sun and Wong, Phys. Rev. A 79, 013824, 2009, and Grange et al, Phys. Rev. Lett. 114, 193601, 2015) therefore refer to this quantity directly as “indistinguishability”. The probability to observe a coincidence detection event between the output ports is in turn reduced to $1-T2/(2T1)$. If an interference experiment is carried out, this reduction in probability corresponds to observing an interference fringe (i.e. a HOM dip) with a visibility of $T2/(2T1)$. A measurement of the visibility can therefore be regarded as a quantification of the indistinguishability or probability for two-photon coalescence.

In actual experiments, additional effects (interferometer imbalance, nonzero single photon purity, or biexponential population decay) will affect the measured coincidence count probability, such that a measurement of the visibility will generally not yield $T2/(2T1)$. Under such conditions, the indistinguishability is frequently quantified by the corrected two-photon interference visibility, which one would expect to measure if the experiment were carried out with single photons in a balanced interferometer. We also choose this method in our analysis.

To avoid confusion in our manuscript, we now simply state that “the indistinguishability can be quantified by $T2/(2T1)$ ” in the introduction, and added another reference which explicitly terms this ratio “indistinguishability” (Sun and Wong, Phys. Rev. A 79, 013824, 2009). When discussing the results of our HOM experiments, we explicitly state that we quantify the indistinguishability by the two-photon interference visibility.

3. Results from different type of cavities (For example, bad cavity) for generation of Purcell enhanced indistinguishable photon is missing.

We thank the Reviewer for this constructive suggestion. To the best of our knowledge, all demonstrations of photon indistinguishability in cavity-based experiments to date have relied

on enhancing the lifetime via the Purcell effect at cryogenic temperatures. The respective quantum emitters in the solid state have been restricted to self-assembled quantum dots, NTDs and Erbium ions. The corresponding experiments employed various cavity geometries, and all were operated either in the regime of coherent (or even strong) coupling or incoherent bad coupling. As suggested by the Reviewer, we have now added to the second paragraph of our introduction references to the related results from different types of cavities (bullseye cavities, open dielectric cavities, micropillar cavities, dielectric ring resonators and photonic crystal cavities and plasmonic nanocavities) for generation of Purcell enhanced indistinguishable photons, and also state explicitly the corresponding regimes of cavity-coupling. The new references also include those suggested by the Reviewer in point 1 above, and we hope our selection reflects the diversity of approaches currently pursued in the field. It is evident from all this previous work, as we state now more clearly, that strong dephasing has to date prevented the demonstration of photon indistinguishability at ambient conditions even in the presence of Purcell enhanced lifetimes. This is because increased dephasing at ambient temperatures will decrease the coherence time and also the Purcell factor, as is evident from the expressions given in our supplemental material. The decrease in Purcell factor could in principle be counteracted by increasing the light-matter coupling strength, e.g. by using plasmonic nanocavities with ultra-small mode volumes, but a respective experimental demonstration has remained elusive to date.

4. For the sake of reproducibility of results, Steps for Fabrication of cavity including DBR coating preparation should be elaborated.

We thank the Reviewer for pointing out the missing information in our description of the cavity. In the Methods section, we have now included two references describing the machining of the fiber tip. The coatings of the fiber tip and the macroscopic mirror were fabricated by a commercial manufacturer (Laseroptik GmbH, Garbsen, Germany) by ion beam sputtering according to the transmission at the design wavelength, which we now state explicitly in the Methods section.

List of changes

All changes made to the manuscript and the Supplementary Information, as detailed in the point-by-point responses above, are highlighted in the revised versions in blue.

Additional changes

1. We have corrected a typo in the caption of Fig. 4.
2. We added explicit statements that the error bars in all figures correspond to the standard uncertainty.
3. Our previous argument for a photon emission efficiency which outperforms spectral filtering was based on a measurement of the saturation count rate. The saturation count rate was measured for an emitter on which no HOM measurement was performed, in order to avoid degradation of NTD 1 and NTD 3. When revising our manuscript, we realized that a lower bound on the efficiency can be obtained from the maximum measured count rate. Computing

this lower bound for NTD 1, we found that the efficiency outperforms that expected for spectral filtering at the same bandwidth by at least a factor of four. In our opinion, this demonstrates the simultaneous enhancement of indistinguishability and photon emission efficiency in our system even more unambiguously than the previous measurement, since the enhancement of both quantities has been experimentally verified on a single emitter. We have revised the relevant paragraphs in our manuscript and the Supplementary Information accordingly, and hope that the Reviewers will also acknowledge this as an improvement.

REVIEWER COMMENTS

Reviewer #1 (Remarks to the Author):

We gratefully acknowledge the authors response to our inquiries. However, considering their argumentation, unfortunately, we need to hold onto our prior assessment and consider the manuscript as not suitable for publication.

1) As we have indicated in our initial review, it is odd, and reader-unfriendly to swap between emitters within the discussion. Especially in Fig 4, where the co/cross measurement is conducted for NTD 3 and the HOM Dip measurement (time delay) is carried out for NTD1, this approach is very concerning. Despite the fact, that we have pointed the attention of the authors to this shortcoming, it was not corrected.

2) What we find even more disturbing, is the fact, that the authors omit to show any other correlation histogram except the one in Fig 4b (which is again plotted in the SI as Fig S1). We believe that this is a serious problem if we consider the level of fluctuations in the correlation peaks. Why do the authors omit to show the histograms, based on which they have extracted the HOM dip of NTD1?

3) The authors have done some effort to discuss the statistical significance of their results in their rebuttal letter. But the fact alone, that the difference in counts of the correlation peak at $\tau=0$ is ~ 10 (peak height), which is smaller than the difference of the correlation peak at -12 ns (~ 20) and of various other peaks in Fig S1 tells us, that the quality of the correlation histogram is simply not good enough to consolidate a 'first -ever' claim in a high impact journal.

4) The Authors argue that the system operates in the incoherent good cavity region: from Ref. 30, this corresponds to low values of light-matter coupling strength (g) and low cavity losses (κ) with respect to the decay rate. In particular, the relation enclosing this region demands $2g \ll \gamma + \gamma^* + \kappa$ and $\kappa < \gamma + \gamma^*$.

The interesting outcome of such regime of interaction is that one can obtain high values of indistinguishability (at room temperature), but tremendously small values of brightness (as

the probability to emit photons in a targeted mode, β factor, tends to zero in this region), see Figs. 2a,b of Ref. 30.

At the end of Ref. 30, it explains that a room temperature quantum dot would require the following parameters to operate in the incoherent good cavity coupling: $g = 120 \mu\text{eV}$, $\kappa = 20 \mu\text{eV}$, $\gamma = 60 \mu\text{eV}$, $\gamma^* = 7000 \mu\text{eV}$. This would render the following values of indistinguishability and β factor: $I = 0.72$, $\beta = 0.088$

The Authors of this work provide the following set of parameters: $g = 21\text{-}80 \mu\text{eV}$ (see below, we believe that there is a mistake with these values), $\kappa = 35.4 \mu\text{eV}$, $\gamma = 4\text{-}0.3 \mu\text{eV}$ (1-15 ns, not measured by the Authors), $\gamma^* = 8000 \mu\text{eV}$

Now, considering the work of L. Husel and coworkers, they present the measurements of the following crucial parameters:

Purcell accelerated lifetime, $\gamma' = 7.3 \mu\text{eV}$ (~ 90 ps), see Fig. 4d. From the main text and supplementary material (in particular, Sec. III), we see that the Authors assume that the non-radiative decay rate is zero $\gamma_{nr}=0$

Total pure dephasing rate ($\Gamma'=\gamma'+2\gamma^*$), measured via scanning the cavity across the emitter spectrum: we are concerned whether this technique offers a correct value of the total dephasing rate, since the scanning cavity changes continuously the detuning with the emitter along the scan. Since they assume that $\gamma^* \gg \gamma'$, then they consider $\Gamma' \sim 2\gamma^*$, and so, the pure dephasing rate is extracted as $\gamma^* = 8000 \mu\text{eV}$, derived from Fig. 2b

The work does not show the measurement of the cavity losses κ , however the Authors indicate that this value is $35.4 \mu\text{eV}$ (to the best of our knowledge, Fig. 2 depicts the spectral position of cavity modes, but not their linewidth).

The Authors infer the light-matter coupling strength (g), via the supplementary Eq. 1, assuming that the radiative decay is 1-15 ns (not measured and referenced to works 3,4 of the supplementary). We believe that they should characterise such crucial parameter entering in the denominator of the Eq. S1, otherwise, the next steps in the calculation of g are uncertain.

The g coupling parameter has a capital relevance to sustain the claim of incoherent good-cavity coupling, we consider that a more rigorous characterisation of such parameter is necessary to defend the indistinguishability values that are measured via two-photon interference and shown in the main text.

If a clear measurement of the two-photon interference would have been presented, a

partial characterisation of the relevant system parameters (as in the current manuscript) would suffice to sustain the claims of the paper, but we believe that the current experiments compiled in this work are incompatible to sustain its publication. If the count rate of the emitter is too low to achieve better HOM measurements, then we encourage the Authors to characterise the coherence time of the system via Michelson interferometry.

Reviewer #3 (Remarks to the Author):

Authors have responded to all my comments and I am satisfied with the answers. They have also improved the manuscript after incorporating the suggested changes.

Cavity-enhanced photon indistinguishability at room temperature and telecom wavelengths

by Lukas Husel et al.

Responses to Reviewer #1

We gratefully acknowledge the authors response to our inquiries. However, considering their argumentation, unfortunately, we need to hold onto our prior assessment and consider the manuscript as not suitable for publication.

We thank the Reviewer for again taking the time and effort to evaluate our manuscript. We have identified the Reviewer's interest in the uncertainties in our HOM measurements as a common ground from which to begin the following responses, thoroughly addressing the concerns raised by the Reviewer.

1) As we have indicated in our initial review, it is odd, and reader-unfriendly to swap between emitters within the discussion. Especially in Fig 4, where the co/cross measurement is conducted for NTD 3 and the HOM Dip measurement (time delay) is carried out for NTD1, this approach is very concerning. Despite the fact, that we have pointed the attention of the authors to this shortcoming, it was not corrected.

Having elaborated in our previous response that the statistical significance of both emitters NTD1 and NTD3 is equivalent, we have refrained from changing the figure. According to our understanding, the presentation of complementary data from multiple emitters and devices in a single figure is common practice and in our case rather strengthens the matter of evidence. This is why we are still reluctant to present in Fig. 4 the data of one emitter only. To address the Reviewer's concern with respect to reader-friendliness nonetheless, we now also show histograms obtained for NTD1 for interferometer delays 0 and 5 ps in Fig. 4. These delay settings are approximately equivalent yet complementary to the co- and cross-polarized interferometer configurations used for NTD3. The additional histograms illustrate the equivalence between the two measurement methods for the interference visibility and guide the reader's eye through the discussion, which we believe improves the presentation of our data. We thank the Reviewer for pointing out this aspect once again.

2) What we find even more disturbing, is the fact, that the authors omit to show any other correlation histogram except the one in Fig 4b (which is again plotted in the SI as Fig S1). We believe that this is a serious problem if we consider the level of fluctuations in the correlation peaks. Why do the authors omit to show the histograms, based on which they have extracted the HOM dip of NTD1?

It was not obvious to us from the first round of reviews that the Reviewer requested the presentation of the histograms measured to extract the HOM dip for NTD1. As elaborated in the previous response, we have now added HOM autocorrelation data of NTD1 for interferometer delays 0 and 5 ps in the revised Fig. 4, and also the raw histograms for these delays as well as for a delay of -5 ps in the new Fig. S2 of the revised Supplementary Information. As mentioned above, switching between these delay settings is equivalent to switching between co- and cross-polarized interferometer configurations, such that the reduction in coincidence counts at zero time delay $\tau = 0$ demonstrates two-photon interference for NTD1.

3) The authors have done some effort to discuss the statistical significance of their results in their rebuttal letter. But the fact alone, that the difference in counts of the correlation peak at $\tau=0$ is ~ 10 (peak height), which is smaller than the difference of the correlation peak at -12 ns (~ 20) and of various other peaks in Fig S1 tells us, that the quality of the correlation histogram is simply not good enough to consolidate a 'first –ever' claim in a high impact journal.

We would like to begin our response by pointing out that the precision and significance of experimental data is independent of the journal it is submitted to. Also, it is not a matter of subjective assessment as “simply not good enough”, but exclusively a question of statistical significance. In the specific context of our work, the analysis is based on the fact that each peak in shot-noise limited correlation histograms is an independent and binomially-distributed random variable, as discussed by Fischer et al. in a thorough theoretical analysis (New J. Phys. 18, 113053, 2016). Accordingly, the method to determine the uncertainty in the type of measurements reported in our work, is to compute the autocorrelation function at zero delay as $g_{HOM}^{(2)}(0) = \frac{N_0}{N_\infty} (1 \pm \sqrt{\frac{1}{N_0} + \frac{1}{N_\infty}})$, where N_0 is the height of the peak at time delay $\tau = 0$ and N_∞ is the mean peak height for $|\tau| > 12.5$ ns. A theoretical derivation of the uncertainty in N_0 is given in Fischer et al. Evidently, the uncertainty in $g_{HOM}^{(2)}(0)$ is dominated by the height of the peak at $\tau = 0$. No other individual peak, such as those selected by the Reviewer, has a significant influence on the computation, since the height only enters via the mean N_∞ . By considering peak-to-peak fluctuations, the Reviewer seems to acknowledge this effect of shot noise on the measurement uncertainty. Taking this as a common ground of understanding, a rigorous quantitative analysis of the set of data underlying our work unambiguously results in the conclusion that our measurements constitute a statistically significant demonstration of two-photon interference.

For completeness, we elaborate on the uncertainties in our measurements, based on the established method referenced above. We obtain $V = 0.65 \pm 0.24$ and $V = 0.51 \pm 0.21$ for the two-photon interference visibilities of NTD1 and NTD3, respectively. The probability that the photons exhibit a statistically significant degree of quantum coherence (i.e. that the value of V is larger than zero) is 99.6% for both emitters individually, providing clear evidence for two-photon interference. The probability that V is two orders of magnitude larger than the expected free-space value is 93% for NTD1, establishing a statistically significant proof for the cavity-enhancement of the indistinguishability. The histogram peaks at $\tau = -12.5$ ns, on which the Reviewer has placed particular emphasis, agree with the expected value of 0.77 ± 0.02 within statistically significant bounds, which we detail in the Supplementary Information.

4) The Authors argue that the system operates in the incoherent good cavity region: from Ref. 30, this corresponds to low values of light-matter coupling strength (g) and low cavity losses (κ) with respect to the decay rate. In particular, the relation enclosing this region demands $2g \ll \gamma + \gamma^* + \kappa$ and $\kappa < \gamma + \gamma^*$.

The interesting outcome of such regime of interaction is that one can obtain high values of indistinguishability (at room temperature), but tremendously small values of brightness (as the probability to emit photons in a targeted mode, β factor, tends to zero in this region), see Figs. 2a,b of Ref. 30.

At the end of Ref. 30, it explains that a room temperature quantum dot would require the following parameters to operate in the incoherent good cavity coupling: $g = 120 \mu\text{eV}$, $\kappa = 20 \mu\text{eV}$, $\gamma = 60 \mu\text{eV}$, $\gamma^* = 7000 \mu\text{eV}$. This would render the following values of indistinguishability and β factor: $I = 0.72$, $\beta = 0.088$

We agree with the Reviewer on the benefit of cavity-enhanced photon indistinguishability and efficiency in the incoherent good cavity regime. As clearly stated in our manuscript and acknowledged by the Reviewer, operation in this regime as defined by Grange et al. (Phys. Rev. Lett. 114, 193601, 2015) requires $2g \ll \gamma + \gamma^* + \kappa$ and $\kappa < \gamma + \gamma^*$ to hold. We point out that the purely hypothetical quantum dot example of Grange et al. mentioned by the Reviewer considers different material and cavity platforms than those used in our work. Of immediate relevance to our work are the actual system parameters, which we determine experimentally and elaborate upon in the following.

The Authors of this work provide the following set of parameters: $g = 21\text{-}80 \mu\text{eV}$ (see below, we believe that there is a mistake with these values), $\kappa = 35.4 \mu\text{eV}$, $\gamma = 4\text{-}0.3 \mu\text{eV}$ (1-15 ns, not measured by the Authors), $\gamma^* = 8000 \mu\text{eV}$

Now, considering the work of L. Husel and coworkers, they present the measurements of the following crucial parameters:

Purcell accelerated lifetime, $\gamma' = 7.3 \mu\text{eV}$ (~90 ps), see Fig. 4d. From the main text and supplementary material (in particular, Sec. III), we see that the Authors assume that the non-radiative decay rate is zero $\gamma_{nr}=0$

Total pure dephasing rate ($\Gamma=\gamma'+2\gamma^*$), measured via scanning the cavity across the emitter spectrum: we are concerned whether this technique offers a correct value of the total dephasing rate, since the scanning cavity changes continuously the detuning with the emitter along the scan. Since they assume that $\gamma^* \gg \gamma'$, then they consider $\Gamma \sim 2\gamma^*$, and so, the pure dephasing rate is extracted as $\gamma^* = 8000 \mu\text{eV}$, derived from Fig. 2b

The work does not show the measurement of the cavity losses κ , however the Authors indicate that this value is $35.4 \mu\text{eV}$ (to the best of our knowledge, Fig. 2 depicts the spectral position of cavity modes, but not their linewidth).

The Authors infer the light-matter coupling strength (g), via the supplementary Eq. 1, assuming that the radiative decay is 1-15 ns (not measured and referenced to works 3,4 of the supplementary). We believe that they should characterise such crucial parameter entering in the denominator of the Eq. S1, otherwise, the next steps in the calculation of g are uncertain. The g coupling parameter has a capital relevance to sustain the claim of incoherent good-cavity coupling, we consider that a more rigorous characterisation of such parameter is necessary to defend the indistinguishability values that are measured via two-photon interference and shown in the main text.

The relevant parameters in our system are as follows:

Cavity linewidth $\kappa = 35.4 \mu\text{eV}$, obtained from the measured transmission of a spectrally narrow laser diode. For the sake of completeness, we now describe this measurement in the revised Supplementary Information.

Total population decay rate of slow population decay component $\gamma = 7.4 \mu\text{eV}$, obtained from the lifetime measurement described in the manuscript. As defined explicitly in Sections I and III of the Supplementary Information, γ is the population decay rate, which together with the separate definition of the radiative decay rate γ_{rad} in Section III implies $\gamma = \gamma_{rad} + \gamma_{nr}$ with the nonradiative decay rate γ_{nr} . Since τ_{rad} is larger than 90 ps (see below), γ is dominated by nonradiative decay processes as $\gamma \approx \gamma_{nr}$. To avoid confusion, we now state $\gamma = \gamma_{rad} + \gamma_{nr}$ explicitly in Section I of the revised Supplementary Information.

Pure dephasing rate $\gamma^* = 8 \pm 2$ meV, obtained from the data shown in Fig. 2b as described in the main text. It is evident from this measurement that the FWHM linewidth Γ of the PL emission profile is 16 ± 4 meV, which is orders of magnitude larger than the measured value $\kappa = 35.4$ μ eV. In this scenario, Γ equals the linewidth of the emitter, which can straightforwardly be inferred from the work by Auffèves et al. (Phys. Rev. A 79, 053838, 2009) as we state in the main text, or alternatively from the explicit expression for the PL emission profile given by Eq. 7 in Meldrum et al. (Optics Express 18, 10, 10230, 2010). As a result, $\Gamma = \gamma + 2\gamma^*$ holds, which, in conjunction with the measured result $\gamma \ll \Gamma = 16$ meV yields γ^* . We also point out that changing the detuning between cavity and emitter is necessary to measure γ^* in this experiment.

Radiative decay rate $\gamma_{rad} = 0.055 - 0.66$ μ eV, corresponding to radiative lifetime $\tau_{rad} = 1 - 15$ ns. These values were obtained from measurements performed on similar NTDs in He et al. (Nat. Photon. 11, 577, 2017) and Hartmann et al. (ACS Nano 10, 8355, 2016). We emphasize that assessing system parameters based on literature values is common practice, especially if well-established emitters such as NTDs are considered. However, to dispel the Reviewer's concern with respect to this parameter and additionally verify our assessment, we determined the radiative lifetime directly from the measured emission efficiency as described in the revised Supplementary Information. Neglecting the fast population decay component, we find an experimentally determined upper bound of $\tau_{rad} = 35.0 \pm 0.9$ ns. We note that the expression for the lifetime was obtained without any assumptions about the implemented regime of light-matter coupling. Including the fast population component, we find $\tau_{rad} = 12.3 \pm 0.3$ ns, in excellent agreement with previous reports in the literature.

Light-matter coupling strength $g = 21 - 80$ μ eV as obtained from the literature values for the radiative lifetime, in agreement with $g = 22.8 \pm 0.6$ μ eV from the measured efficiency. We note that g also depends on the mode volume, calculated from the measured mode order. The agreement between model and data in Fig. 2c confirms the accuracy of this calculation.

It is obvious that $2g \ll \gamma + \gamma^* + \kappa$ and $\kappa < \gamma + \gamma^*$ hold for the above parameters. Moreover, we note that operation outside the incoherent good cavity regime due to a larger coupling strength, i.e. $2g \gtrsim \gamma + \gamma^* + \kappa$, would require quantum yields on the order of 70%. To the best of our knowledge, all experiments performed on NTDs, including those synthesized by our method, have determined one order of magnitude smaller quantum yields. The rigorous characterization of our system, based on measured quantities and in full agreement with previous experiments on similar cavity setups or emitters, unequivocally places our system into the incoherent good cavity regime.

If a clear measurement of the two-photon interference would have been presented, a partial characterisation of the relevant system parameters (as in the current manuscript) would suffice to sustain the claims of the paper, but we believe that the current experiments compiled in this work are incompatible to sustain its publication. If the count rate of the emitter is too low to achieve better HOM measurements, then we encourage the Authors to characterise the coherence time of the system via Michelson interferometry.

As elaborated above, all relevant parameters in support of all conclusions have been determined explicitly and with statistical significance. Therefore, there remain no physical quantitative arguments to question our work. The rigorous analysis of all relevant system parameters constitutes an unambiguous experimental demonstration of operation of our system in the regime of incoherent good cavity coupling. We therefore anticipate that in the current round of revision, the Reviewer will acknowledge the quantitative and statistically significant aspects of our work and support its publication.

List of changes

All changes made to the manuscript and the Supplementary Information are highlighted in the revised versions in blue.

Additional changes

1. We have corrected a typo in Supplementary Section IIIB.
2. We have corrected a typo in the definition of the ideal Purcell factor in Supplementary Section IIIA. The calculation of this quantity was also off by a factor of π , which we have now corrected. We emphasize that these corrections have no impact on the results discussed in our manuscript, as the ideal Purcell factor is of no relevance to the analysis.
3. We have reworded two expressions in the discussion of the Purcell effect in the main text to condense the manuscript.
4. Titles were added to the Supplementary Fig. S1 and S2 to differentiate between the two emitters NTD3 and NTD1.

REVIEWERS' COMMENTS

Reviewer #1 (Remarks to the Author):

We thank the authors for their efforts to account for our comments and suggestions.

We believe, that the inclusion of more data, both in the manuscript and the SI substantially strengthens the claim of observed indistinguishability. Also, providing the details on the data analyses greatly improves the transparency and readability.

Under these circumstances, we believe that the paper can be published, in principle.

However, we would encourage the authors to add even more correlation histograms in the SI Fig S2 for Sample NTD1, e.g. at ± 2 ps delay, to really demonstrate the consistency of the growing central peak in the correlation histogram.

Manuscript NCOMMS-23-33232B

Cavity-enhanced photon indistinguishability at room temperature and telecom wavelengths
by Lukas Husel et al.

Responses to Reviewer #1

We thank the authors for their efforts to account for our comments and suggestions. We believe, that the inclusion of more data, both in the manuscript and the SI substantially strengthens the claim of observed indistinguishability. Also, providing the details on the data analyses greatly improves the transparency and readability. Under these circumstances, we believe that the paper can be published, in principle.

We thank the reviewer for the positive assessment of our revised work and the recommendation for publication.

However, we would encourage the authors to add even more correlation histograms in the SI Fig S2 for Sample NTD1, e.g. at +/- 2 ps delay, to really demonstrate the consistency of the growing central peak in the correlation histogram.

We fully agree with the reviewer on this matter. In Supplementary Figure S2 we now include five histograms in intervals that are spaced approximately evenly over the full range of interferometer delays in order to demonstrate the consistency of the central peak decreasing towards zero delay.

List of changes

All changes made to the manuscript and Supplementary Information are highlighted in blue.

Additional changes

1. We noticed an error in the expression for the light-matter coupling strength g in Supplementary Eqn. 1, resulting in a 20% correction to the originally stated upper bound for g . As a result, we have also revised and corrected our estimates for the quantities R , F_{p^*} and the quantum yield in Supplementary Sections I, II and III.A (see also point 3 below). All other claims and statements in the manuscript remain unaffected by these changes.
2. Triggered by the editor's summary and by the feedback of colleagues from the field, we realized that we have not conveyed the mechanism of Purcell-enhanced photon emission in our system clearly enough. We operate in a regime where the emitter's decay rate is not enhanced by the cavity, since the emitter's spectral width greatly exceeds that of the cavity. However, the small fraction of the emitter's spectrum that couples to the cavity experiences full Purcell enhancement, resulting in an

enhancement of the emission spectral density which is quantified by the ideal Purcell factor. This mechanism, which is clearly illustrated in Kaupp et al, Phys. Rev. A 88, 053812 (2013), is the reason why the single photon efficiency in our system exceeds that expected for spectrally filtered free-space NTD emission. To elaborate this point, we now explicitly refer to the emission spectral density in the first sentence of paragraph 4 in the main text, and added a short explanation to Supplementary Sec. III.B. We have also revised the heading and first sentence of Supplementary Sec. III.C to avoid any potential confusion with experiments operating in regimes of large Purcell enhancement, where the population lifetimes are shortened by the cavity. We emphasize that these additions and changes merely serve to improve the clarity of the presentation without any consequence for the original claims of the manuscript.

3. When revisiting the role of Purcell enhancement in our system, we realized that our previous estimate for the enhancement of the emission spectral density, i.e. the factor by which we outperform the spectral filtering method, was extremely conservative. Similar to the theoretical proposal by Grange et al, Phys. Rev. Lett. 114, 193601 (2015), we had used an upper bound for the efficiency expected for spectral or temporal filtering of free-space NTD emission. In our system, the expected NTD quantum yield is drastically smaller than unity (see also point 1 above), such that we expect the efficiency for the free-space filtering method to be at least an order of magnitude lower than the given upper bound. We now mention this effect in the second-to-last paragraph of the main text, and explain it elaborately in Supplementary Sec. III.B, alongside the changes outlined in point 2 above. These changes and additions serve to illustrate the benefits of the incoherent good cavity-coupling regime more clearly.